# A Lagrangian model for drifting ecosystems reveals heterogeneity-driven enhancement of marine plankton blooms

Enrico Ser-Giacomi [1,2] ✉, Ricardo Martinez-Garcia[3,4], Stephanie Dutkiewicz[1] & Michael J. Follows[1]

Marine plankton play a crucial role in carbon storage, global climate, and ecosystem function. Planktonic ecosystems are embedded in patches of water that are continuously moving, stretching, and diluting. These processes drive inhomegeneities on a range of scales, with implications for the integrated ecosystem properties, but are hard to characterize. We present a theoretical framework that accounts for all these aspects; tracking the water patch hosting a drifting ecosystem along with its physical, environmental, and biochemical features. The theory resolves patch dilution and internal physical mixing as a function of oceanic strain and diffusion. Ecological dynamics are para-meterized by an idealized nutrient and phytoplankton population and we specifically capture the time evolution of the biochemical spatial variances to represent within-patch heterogeneity. We find that, depending only on the physical processes to which the water patch is subjected, the plankton biomass response to a resource perturbation can vary in size up to six times. This work indicates that we must account for these processes when interpreting and modeling marine ecosystems and provides a framework with which to do so.

Plankton blooms in the ocean represent some of the most massive and rapid biomass growth events in nature. Planktonic organisms are the base of the marine food chain, contribute to the cycling of carbon, and preserve ocean biodiversity[1,2]. Notably, phytoplankton blooms are not uniformly distributed across the seascape. The large spatio-temporal scales of phytoplankton distribution are set by seasons and basin-wide circulation. On a smaller scale, eddies[3] and fronts[4,5] contort these patterns, and localized injections of nutrients into the sunlit layer allow for the formation of frequent and ephemeral blooms (e.g., as seen in satellite observation, Fig. 1). Such pulses of resources could be caused, for instance, by upwelling of nutrient-rich water[3], a burst of micro-nutrients from dust deposition[6], the wake of islands[7], or by deliberate fertilization experiments as have been carried out in several location in

the ocean[8,9]. The rich structure in observed chlorophyll at those scales demands tools for interpretation. How do such bloom events evolve as a result of the local bio-physical environment?

Once favorable conditions for growth are set, the fate of a plankton ecosystem is indeed tightly linked to the physical evolution of the patch of water that contains it. Here we define a patch as a physical body of ocean water identified by precise position, shape and size at a given time. Thus, the interplay of strain and diffusion gener-ated by oceanic currents can strongly deform, dilute and mix a water patch and such processes could affect the associated ecosystem in various ways[10–14]. Dilution has been proposed as a prominent driver of plankton productivity by modulating concentrations of nutrients and biomass within a patch of water[9,15–17]. This has been associated either

[1]Department of Earth, Atmospheric and Planetary Sciences, Massachusetts Institute of Technology, 54-1514 MIT, Cambridge, MA 02139, USA. [2]IFISC (CSIC-UIB), Institute for Cross-Disciplinary Physics and Complex Systems, Palma de Mallorca, Spain. [3]ICTP South American Institute for Fundamental Research & Institute of Theoretical Physics, Universidade Estadual Paulista - UNESP, Rua Dr.Bento Teobaldo Ferraz 271, Bloco 2 - Barra Funda, 01140-070 São Paulo, SP, Brazil. [4]Center for Advanced Systems Understanding (CASUS); Helmholtz-Zentrum Dresden-Rossendorf (HZDR), Görlitz, Germany. ✉e-mail: enrico.sergiacomi@gmail.com

with a condition of stoichiometric unbalance where a nutrient inside a bloom becomes suddenly limiting[9,15,18] or with the effect of diluting grazer concentrations in a region where they are higher than average[16]. On the other hand, the high level of spatial heterogeneity - i.e., patchiness - generated by ocean turbulence across a wide range of scales can also potentially affect biomass production[19–23]. Indeed, due to the often non-linear response of plankton to nutrients, biomass growth can depend not only on average concentrations of resources but also on their spatial heterogeneity. Thus, a mechanistic understanding of the dynamics of plankton ecosystems in its entirety requires a Lagrangian approach - that is following the water patch within which plankton live. At the same time, the role of spatial heterogeneity inside such dynamic ecosystems should be carefully addressed.

However, the combined impact of the Lagrangian evolution of a water patch and the associated spatial heterogeneity as aspects of the same evolving system has not yet been addressed. Lagrangian models to date always assume a water parcel is well-mixed, i.e. spatially homogeneous[13,16,17,24], and patchiness of quantities such as nutrients or biomass has never been implemented in a Lagrangian frame of reference[25–28]. By combining both we will be better able to disentangle the physical from the biological drivers of the generation, maintenance and decay of plankton blooms in the ocean[5].

Here we introduce a new framework to track, from first principles, a generic plankton ecosystem from a Lagrangian perspective. We define a Lagrangian patch as a physical body of water of arbitrary shape and size containing such an ecosystem. We study the physical dynamics, the evolution of spatial heterogeneity within the patch as well as the biochemical interactions between nutrients and their consumers. Though the theoretical approach we develop could be used in many applications, we concentrate on the ecological response to pulses of resources within such Lagrangian ecosystems while they are subjected to dilution with its resource-poorer surroundings. As first application, we model the biophysical evolution of the artificially-fertilized bloom during the SOIREE campaign obtaining predictions consistent with the observed data[13,29]. More generally, we then demonstrate that dilution, driven by strain and diffusion, is responsible for the initial generation of patchiness in Lagrangian plankton ecosystems. Finally, we show that such heterogeneity can in turn significantly enhance plankton growth highlighting the existence of optimal dilution rates that maximize the patch-integrated biomass.

## Results

### Lagrangian ecosystems theory

We develop a theoretical framework to study a generic plankton ecosystem inhabiting a Lagrangian patch of water at the ocean surface. The Lagrangian perspective - that is tracking in space and time the same physical water mass - allows us to naturally address the ecological responses to favorable (or unfavorable) environmental conditions occurring in the patch itself (Fig. 1). In this section we layout all the essential concepts and quantities to describe our approach; the mathematical developments are extensively illustrated in the Methods. Model variables are listed in Table 1. We first focus on the physical evolution of the patch and then we describe the associated tracers dynamics.

Any Lagrangian water patch in the ocean associated with an initial physical body of fluid, undergoes continuous changes in position, size and shape due to the effect of ocean motions (Fig. 2). To model the physical transformations of the patch, we approximate it by an ellipse containing a prescribed percentage of its surface[11,30]. This allows to track the evolution of the same mass of fluid in time along with its dilution and mixing with surrounding waters. The patch shape is thus described, at time $t$, by the length $L(t)$ and the width $W(t)$ of such ellipse. Its characteristic size is defined as $S(t) = L(t) + W(t)$ while its area is $A(t) = \pi W(t) L(t)$ (Methods). From a Lagrangian perspective all rigid-like movements associated with the patch, such as translation and rotation, are ignored because they are implicitly incorporated in the displacement of the frame of refs. [31–33]. Previous studies have shown that a water patch in the open ocean is primarily affected by horizontal strain and diffusion[11,30]. The strain rate $\gamma(t)$ is responsible for the elongation of the patch, augmenting its aspect ratio. Diffusion $\kappa(t)$ describes the small-scale processes that cause the entrainment of surrounding waters within the patch. With the addition of water into the patch, its area increases (Fig. 3). Solving a Lagrangian advection-diffusion equation[10,11,21,30,32,34,35] we obtain analytical expressions for the evolution of $W(t)$ and $L(t)$ and from them we derive the patch area increase rate as function of $\gamma(t)$ and $\kappa(t)$ (Methods):

$$\frac{dA(t)}{dt} = \pi\kappa(t)\left[\frac{W^2(t)+L^2(t)}{W(t)L(t)}\right]. \tag{1}$$

From Eq. (1) we see that diffusion has a stronger proportional effect on the area increase when the perimeter-to-area ratio of the patch is larger. The strain rate controls how fast this ratio increases. Indeed the quantity $W^2(t) + L^2(t)$ is proportional to the square of the perimeter of the ellipse encompassing the patch and thus $\frac{dA(t)}{dt} \propto \frac{\kappa(t)(\text{perimeter})^2}{\text{area}}$. Therefore, even though strain does not directly contribute to mix the patch with the surrounding, it makes diffusion more efficient by

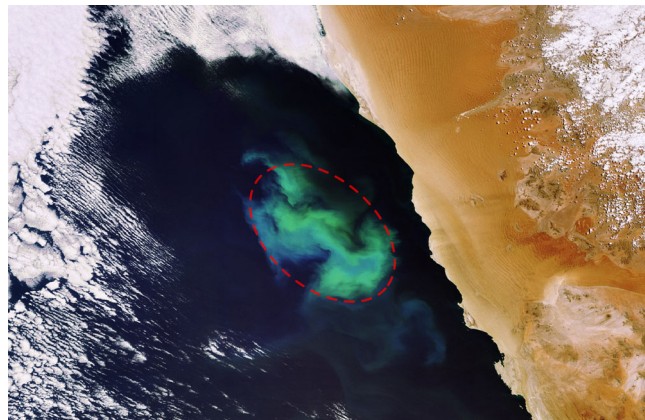

**Fig. 1 | A plankton bloom offshore of Namibia.** The picture was taken by the MERIS (Medium Resolution Imaging Spectrometer) instrument aboard ESA's Envisat satellite on 6/11/2007. The red dashed elliptical line indicates the water patch where the bloom is occurring. Coupled biophysical processes lead to a marked spatial heterogeneity - i.e patchiness - within the region.

**Table 1 | Symbols, names and units of the model variables**

| Variable | Name | Units |
|---|---|---|
| $L, W, S$ | Length, width and size of the patch | $km$ |
| $\gamma$ | Strain rate | $1/day$ |
| $\kappa$ | Diffusion | $km^2/day$ |
| $p_i$ | Patch concentration of $i$-tracer | $\mu mol/m^3$ |
| $s_i$ | Surrounding concentration of $i$-tracer | $\mu mol/m^3$ |
| $v$ | Maximum growth rate | $day^{-1}$ |
| $\alpha$ | Remineralization fraction | – |
| $m$ | Mortality rate | $day^{-1}$ |
| $k$ | Half-saturation constant | $\mu mol/m^3$ |
| $\tau$ | Integration time | $day$ |
| LBA | Lagrangian biomass anomaly | $MgC$ |

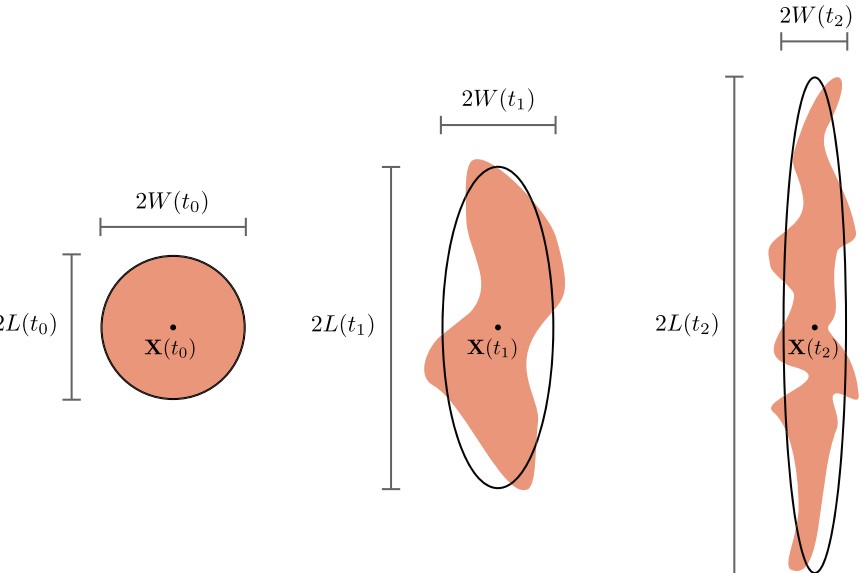

**Fig. 2 | The transformation of the shape of a 2-dimensional Lagrangian patch (salmon color) for three consecutive times $t_0 < t_1 < t_2$.** Conceptually, we can can think of these images as if we had dyed all the water molecules in the left "patch" at $t_0$ and then watched is evolution over time. The patch is modeled as an ellipse (black line) with a co-moving center of mass $\mathbf{X}(t)$ and changing characteristic length $L(t)$ and width $W(t)$. The characteristic size of the patch is $S(t) = L(t) + W(t)$ while its area is $A(t) = \pi W(t) L(t)$ (Methods).

increasing the patch perimeter. Consequently, the dilution rate can be defined as the rate at which surrounding waters are entrained in the patch and can be identified with the area increase rate of Eq. (1).

Experimental measurements have shown that, due to the complexity of ocean turbulence, strain, and diffusion values change depending on the spatial scale considered[10,11,30,36–39]. Hence, while a Lagrangian patch is expanding, it can be subject to a range of different values of strain and diffusion. To describe this effect we make the strain and diffusion rates depend on patch size (Methods):

$$\gamma(t) = f_\gamma[S(t)], \tag{2}$$

$$\kappa(t) = f_\kappa[S(t)]. \tag{3}$$

This allows us to describe the patch evolution across different dynamical regimes in the ocean, from sub-meso to gyre scales, matching the corresponding strain and diffusion functional forms[10,38]. This approach permits us to recreate physical patch dynamics observed in the real ocean, such as the decrease and successive increase of the patch width $W$, that cannot be modeled assuming fixed strain and diffusion values.

To characterize the plankton ecosystem associated with a Lagrangian patch, we need to describe its drifting components (i.e. resources and organisms) - generally referred as tracers - in terms of their spatial distributions. Due to diffusive processes at the patch boundaries and its consequent increase in area and dilution, tracers inside the patch will interact and mingle with tracers at the patch surrounding[14]. To model such dynamics explicitly, the inside and outside distributions of tracers have to be described separately. Formally, for a generic tracer $i$, its distribution fields (in terms of, for instance, abundance or mass) within the patch and at its surrounding are denoted as $p_i(\mathbf{x}, t)$ and $s_i(\mathbf{x}, t)$, respectively. Since the tracer fields are not uniform across the ocean, we use the Reynold's decomposition to account for spatial heterogeneity[25–28,40]:

$$p_i(\mathbf{x}, t) = \langle p_i(\mathbf{x}, t) \rangle + p_i'(\mathbf{x}, t) \quad ; \quad s_i(\mathbf{x}, t) = \langle s_i(\mathbf{x}, t) \rangle + s_i'(\mathbf{x}, t) \tag{4}$$

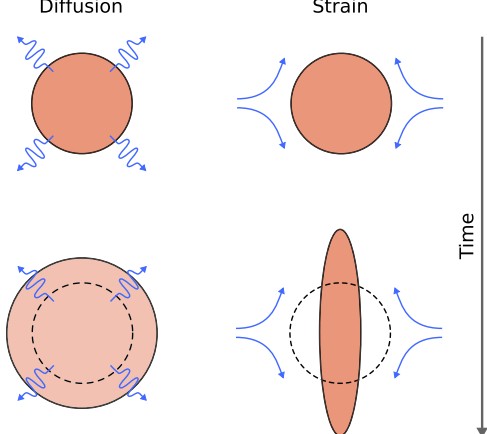

**Fig. 3 | Strain and diffusion effects (blue lines) on an initially circular Lagrangian patch (salmon color).** From top to bottom, the change in time of the patch is sketched. Diffusion (left side) isotropically dilutes the patch through small-scale mixing occurring at its boundary. Strain (right side) stretches the patch conserving its area by increasing its length and compressing its width. Considered together, strain and diffusion generate a wide range of possible combinations of patch shapes and sizes.

where $\langle p_i(\mathbf{x}, t) \rangle$ and $\langle s_i(\mathbf{x}, t) \rangle$ are spatial means while $p_i'(\mathbf{x}, t)$ and $s_i'(\mathbf{x}, t)$ are fluctuations. Thus, second moments - that are spatial variances and covariances - are denoted as $\langle p_i'(\mathbf{x}, t)^2 \rangle$, $\langle s_i'(\mathbf{x}, t)^2 \rangle$ and $\langle p_i'(\mathbf{x}, t) p_j'(\mathbf{x}, t) \rangle$, $\langle s_i'(\mathbf{x}, t) s_j'(\mathbf{x}, t) \rangle$ for any tracer $i$ and $j$ (Fig. 4). We identify the three main determinants of the evolution of tracer fields inside the patch as:

- Entrainment of surrounding waters
- Internal mixing
- Biochemical interactions

Entrainment is intimately related with the patch dilution that, in turn, can be modeled in terms of the patch area increase. We derive general analytical expressions to quantify the effect of such process on the derivative of first (spatial means) and second (spatial variances and covariances) moments of the tracer distributions (Methods and

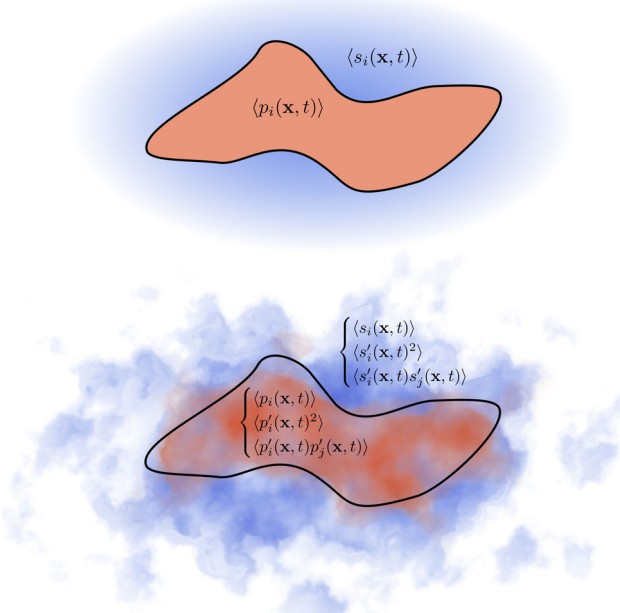

**Fig. 4 | Tracers distributions in the patch (salmon color) and at its surrounding waters (blue color).** When the assumption of well-mixed concentrations is taken (top panel), for a given tracer $i$, we need to specify only its mean concentration in the patch $\langle p_i(\mathbf{x}, t)\rangle$ and at the surrounding $\langle s_i(\mathbf{x}, t)\rangle$. Instead, if we account for spatial heterogeneity (bottom panel), second moments of tracers distributions have to be considered, both in the patch $\langle p_i'(\mathbf{x},t) p_j'(\mathbf{x},t)\rangle$ and at the surrounding $\langle s_i'(\mathbf{x},t) s_j'(\mathbf{x},t)\rangle$, for any $i$ and $j$.

Supplementary Fig. 1):

$$\frac{d\langle p_i(\mathbf{x}, t)\rangle}{dt} = \frac{dA(t)}{dt} \frac{\langle s_i(\mathbf{x}, t)\rangle - \langle p_i(\mathbf{x}, t)\rangle}{A(t)}, \tag{5}$$

$$\frac{d\langle p_i'(\mathbf{x},t) p_j'(\mathbf{x},t)\rangle}{dt} = \frac{dA(t)}{dt} \frac{\left[\langle s_i'(\mathbf{x},t) s_j'(\mathbf{x},t)\rangle - \langle p_i'(\mathbf{x},t) p_j'(\mathbf{x},t)\rangle\right] + \left[\langle s_i(\mathbf{x},t)\rangle - \langle p_i(\mathbf{x},t)\rangle\right]\left[\langle s_j(\mathbf{x},t)\rangle - \langle p_j(\mathbf{x},t)\rangle\right]}{A(t)}. \tag{6}$$

Note that, while the evolution of first moments depends only on the difference between means, the derivatives of second moments are a function of means, variances and covariance.

Internal mixing, on the other hand, is driven by the diffusion of tracers within the patch. Such process reduces the spatial variances and covariances of the tracer fields, but leaves the spatial means unchanged. We model it assuming an exponential decay for variances and covariances[35,41–43]:

$$\langle p_i'(\mathbf{x}, t) p_j'(\mathbf{x}, t)\rangle \sim e^{-\frac{\kappa(t)}{S(t)^2}t}, \tag{7}$$

where $\frac{\kappa(t)}{S(t)^2}$ is the effective decay rate associated with the diffusion $\kappa(t)$ at the spatial scale $S(t)$ (Methods).

Biochemical interactions among tracers within the patch can be addressed using first and second moments of the associated distributions $p_i(\mathbf{x},t)$'s. Due to the modularity of the approach proposed here, different models involving any number of tracers can be implemented, from resource-consumer to niche or neutral models[24,44–50].

External factors such as temperature or light can also be directly included as modulators of biochemical dynamics.

Entrainment, mixing, and interactions are thus the three fundamental actors that shape the spatial distribution of tracers within a Lagrangian plankton ecosystem. Synthesizing the above developments, we can write a master equation for the time evolution of a generic tracer distribution $p_i(\mathbf{x}, t)$ encompassing such physical and biochemical processes:

$$\frac{dp_i(\mathbf{x},t)}{dt} = \mathcal{E}\left[p_i(\mathbf{x},t); s_i(\mathbf{x},t)\right] + \mathcal{M}\left[p_i(\mathbf{x},t)\right] + \sum_j \mathcal{I}_{ij}\left[p_i(\mathbf{x},t); p_j(\mathbf{x},t)\right], \tag{8}$$

where $\mathcal{E}$ includes the contribution of entrainment from Eqs. (5)-(6)), $\mathcal{M}$ the effect of internal mixing from Eq. (7) and $\mathcal{I}_{ij}$ the interactions between tracer $i$ and $j$ which can have different functional forms depending on the dynamics considered. By virtue of the generality of Eq. (8), our framework can be adapted to different spatio-temporal scale while focusing on various physical and biochemical dynamics.

## Modeling a fertilized patch: setup and ensemble simulations
We simulate the dynamics of a Lagrangian plankton ecosystem by integrating Eq. (8). As prototypical approach we model for 30 days an ecosystem initially residing in a 10 km wide and 10 meters thick circular patch (model sensitivity shown in Supplementary Fig. 3). This setting encompasses relevant spatio-temporal scales typical of natural as well as artificial fertilized blooms[8,9,16,22,51]. Functional forms $f_\gamma$ and $f_\kappa$ for the scaling-laws of strain and diffusion are chosen to match their experimentally measured values at the specific spatial scales spanned by the patch size evolution, that are of the order of 10–100 km (Methods).

To address the response of a Lagrangian ecosystem to localized conditions favoring population growth, we focus on the biochemical interactions between two ideal tracers: an inorganic resource $p_r \equiv p_r(\mathbf{x}, t)$ and a planktonic consumer $p_b \equiv p_b(\mathbf{x}, t)$ nourished by the resource. We assume a Monod kinetics and a linear mortality rate for the consumer[16,44,45]. Hence, the general term $\mathcal{I}_{ij}[p_i(\mathbf{x},t); p_j(\mathbf{x},t)]$ of Eq. (8) can be made explicit:

$$\frac{dp_r}{dt} = -\nu \frac{p_r}{p_r + k} p_b + \alpha m p_b, \tag{9}$$

$$\frac{dp_b}{dt} = +\nu \frac{p_r}{p_r + k} p_b - m p_b, \tag{10}$$

where $\nu$ is the maximum growth rate, $k$ is the half-saturation constant, $m$ the linear mortality rate and $\alpha$ is the fraction of dead biomass that is recycled into the resource pool. Accordingly, the biomass "export" rate out of the patch corresponds to $m(1-\alpha)p_b$. Following Eq. (4), we can use the Reynold's decomposition[27,28] to evaluate the contribution of first and second moments to Eqs. (9) and (10) (Methods).

We stimulate Lagrangian blooms of consumer $p_b$ by fertilizing the patch with a pulse of resources that mimics, for instance, processes like nutrient upheaval, dust deposition, fertilization experiments or,

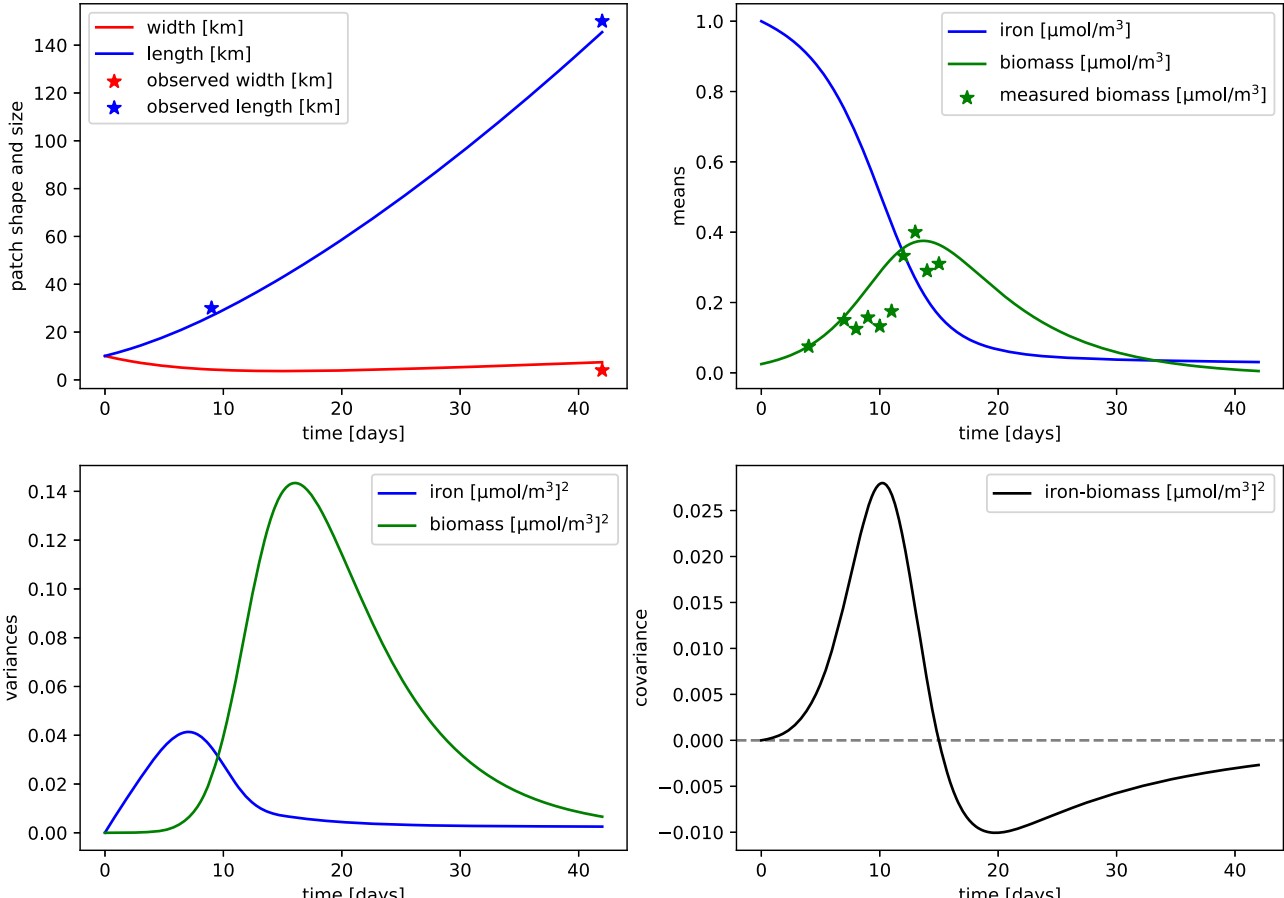

**Fig. 5 | Modeled bio-physical dynamics of the iron-fertilized bloom during the SOIREE experiment.** Initial values of iron and biomass concentrations as well as biological model parameters are set to resemble the ones measured during the campaign (Methods). Solid line represent the time evolution of model variables: patch width and length (top-left), iron and biomass spatial means (top-right), iron and biomass spatial variances (bottom-left) and iron-biomass covariance (bottom-right). Stars corresponds to measured values for the corresponding variables from to in-situ sampling or remote-sensing. Biomass is expressed in iron currency for visualization convenience.

more generally, any perturbation to the average ocean state that causes a local inhomogeneity of resource concentrations. To this aim, we initially (at $t = 0$) fix the second moments to zero and the first moments at steady state, internally to the patch and at its surroundings. This corresponds to an initial state where everything is well-mixed and spatial means of tracer distributions are stationary. Then, we initialize each numerical experiment by increasing the resource mean in the patch $\langle p_r \rangle$ and tracking the ecosystem response. To provide a concrete and realistic interpretation of the model outputs, we set Eqs. (9) and (10) to recreate an idealized iron-phytoplankton dynamics (Methods), adopting values for the biological parameters derived from literature[15,29,44,52–54] (model sensitivity shown in Supplementary Figs. 3, 4, 5 and 8).

As first application, we simulate the iron-fertilized bloom during the SOIREE experiment[29] in the Southern Ocean (see Fig. 5). We focus on this campaign since it is, to our knowledge, the one for which we have the most detailed description of the physical evolution of the water patch hosting the bloom. For this specific simulation we use initial values for strain and diffusion of $\gamma = 0.12\ day^{-1}$ and $\kappa = 0.1\ km^2/day$. In this way the width and length of the modeled patch match satellite observations of the SOIREE bloom taken at 9 and 42 days[13]. The resource $r$ represents iron and the consumer $b$ is a generic phytoplankton type. In accord with experimental data and dedicated models, the simulated bloom that we recreate peaks after 14 days and it reaches values ~ 15 times higher than the surrounding biomass concentration[18,29]. The mean biomass curve predicted by the model follows the in-situ measures taken during the

first 15 days of the campaign. We also conducted sensitivity experiments to show that other strain and diffusion combinations do not well capture the observations (Supplementary Fig. 2), suggesting a tight relation between the physical and ecological evolution of the SOIREE bloom. Thus, despite the simplicity of the biochemical dynamics considered, our model is able to reproduce the main bio-physical patterns of a plankton bloom and demonstrates the key role of dilution.

We also provide the time evolution of second moments of tracers distributions even though they were not measured during the SOIREE experiment. Biomass variance peaks about 10 days after the iron variance and it reaches higher values. This is consistent with observations showing that the plankton distributions are more patchy than the nutrient ones[19,20]. The covariance curve unveils how, for high-resource concentrations, biomass and iron are spatially correlated while, when the resource starts to be depleted, the correlation becomes negative. This inversion almost coincides with the mean biomass peak. This suggests the existence of a dynamical relationship between covariance, and more generally of tracer heterogeneity, and biomass growth[21,27,28,35]. However, the relative contribution of physical versus intrinsic biochemical factors in generating spatial heterogeneity still remains implicit.

To unveil the interrelation between physical forcings and bloom dynamics we produce several ensembles of simulations (Methods). Within each ensemble, we explore ranges of strain and diffusion values maintaining the same initial input of the nutrient. Such combinations of parameters allow us to explore a wide spectrum of patch

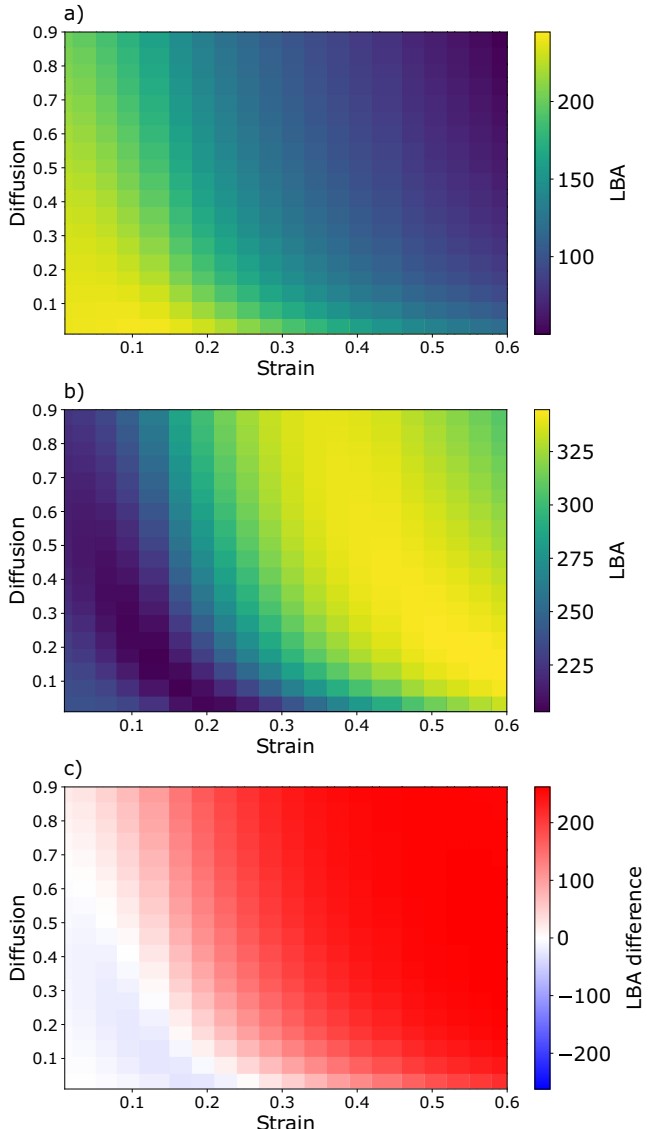

**Fig. 6 | Lagrangian biomass anomaly from ensemble simulations.** We show the Lagrangian biomass anomaly (LBA) measured in tonnes of carbon [Mg C] for an ensemble of well-mixed (**a**) and spatially-heterogeneous (**b**) patches across realistic ranges of strain [day⁻¹] and diffusion [km²day⁻¹] for one month of integration. Panel (**c**) shows the LBA difference between heterogeneous and well-mixed ensembles. The values reported for strain and diffusion are referred to the initial time of the simulations, as the patch size increases, they change accordingly to the respective scaling laws. All the other model parameters are kept constant (Methods). Each pixel of the heatmaps corresponds to the LBA attained by a single simulated patch under the effect of a specific combination of strain and diffusion. In (**a**) spatial heterogeneity is neglected by switching off the second moments of the tracer distributions. The maximum LBA values are reached for the minimums of both strain and diffusion i.e. for the smaller dilution rates. Instead, in (**b**), spatial heterogeneity is explicitly considered by modeling the second moments of the tracer distributions. The LBA in this case reaches much higher values than in the well mixed case as explicitly shown in (**c**). LBA maxima for the heterogeneous case populate an extended ridge in the LBA surface highlighting the fact that the associated optimal dilution values can be obtained from various combinations of strain and diffusion.

dilution rates. In order compare the differences in response between well-mixed patches and heterogeneous patches, we confront ensembles where second moments are switched off with ones where they are fully considered. In well-mixed ensembles variances and covariances are thus always set to zero while in the heterogeneous

ensembles they are free to vary. We also perform independent simulations exploring the model sensitivity and robustness (Supplementary Figs. 3, 4, 5, 7, and 8).

We then introduce a synthetic metric to be able to compare different simulations within ensembles. In particular, we aim at characterizing the overall response of a Lagrangian ecosystem to a resource perturbation with respect to its steady state. We measure such deviation by defining a quantity called Lagrangian biomass anomaly (LBA):

$$\text{LBA} = \frac{1}{\tau} \int_0^\tau A(t)\big(\langle p_b \rangle - \langle s_b \rangle\big) dt. \tag{11}$$

The above expression is the average over the time window $\tau$ of the anomaly of biomass residing in the patch with respect to the surrounding value $\langle s_b \rangle$. Indeed, for any time $t$, the term $A(t)\big(\langle p_b \rangle - \langle s_b \rangle\big)$ is the difference between the absolute biomass in the patch and the biomass of a region of the surrounding of the same area $A(t)$. If LBA > 0, the patch biomass has been on average higher than the surrounding and the opposite if LBA < 0. Hence, the LBA is based on biomass standing stock (potentially evaluated with Chlorophyll) and so provides a useful real-world metric which could be based on remote-sensing. In the model the LBA is also a proxy for the biomass export; combining Eqs. (9), (10) and (11), the temporal mean of the export rate anomaly turns out to be $m(1 - \alpha)$LBA.

## Dilution and spatial heterogeneity trade-offs in enhancing Lagrangian ecosystem biomass

We start by considering the ensemble where patches are forced to be well-mixed, as this is the usual assumption in Lagrangian studies as well as inside grid cells of Eulerian models. We first note that the LBA is always positive, meaning that any fertilized patch produced more biomass than the surrounding (see Fig. 6). However, higher dilution - driven by stronger strain and diffusion - leads to lower LBA respect to low dilution regimes. This might be what we intuitively expect: in a well-mixed Lagrangian ecosystem, the modification of patch mean concentrations described by Eq. (5) due to the entrainment of resource-poorer water always reduces biomass production with respect to the case of a "closed patch" with no exchanges with the surroundings. In other words, any intrusion of surrounding water from outside of a well-mixed patch leads to less increase of biomass than if there was no external water entrained.

If instead we consider the more realistic case where the patch is spatially heterogeneous, Eq. (6) shows that dilution by itself, associated with the intrusion of external water with different tracer concentrations, can generate spatial heterogeneity. In this scenario, our ensemble simulations reveal the existence of a region in the strain-diffusion parameter space in which the LBA is maximal (see Fig. 6). We conclude that dilution-driven spatial heterogeneity could greatly enhance the biomass of a plankton ecosystem. To further support this, in Fig. 7 we plot, for the two ensembles, the LBA versus the average dilution factor - that is ratio between the temporal mean of the patch area and its initial value. In the well-mixed case, the LBA decreases monotonically with dilution and its values can be up to 6 times smaller than in the heterogeneous ensemble, which instead presents a more complex pattern with a marked LBA peak at intermediate dilutions.

Figures 6 and 7 show that the LBA of the heterogeneous ensemble is very similar to the well-mixed one for small dilution values. However, for the heterogeneous case, after touching a minimum valley, the LBA surface rises steadily until reaching a maximum ridge. To investigate such behavior, we calculate the contribution of spatial heterogeneity to biomass production by subtracting all first-moment terms to Eq. (33) and integrating in time. We find that this quantity is below (or close to) zero in the decreasing part of the LBA surface and becomes positive when the LBA begins to rise after reaching its minimum values

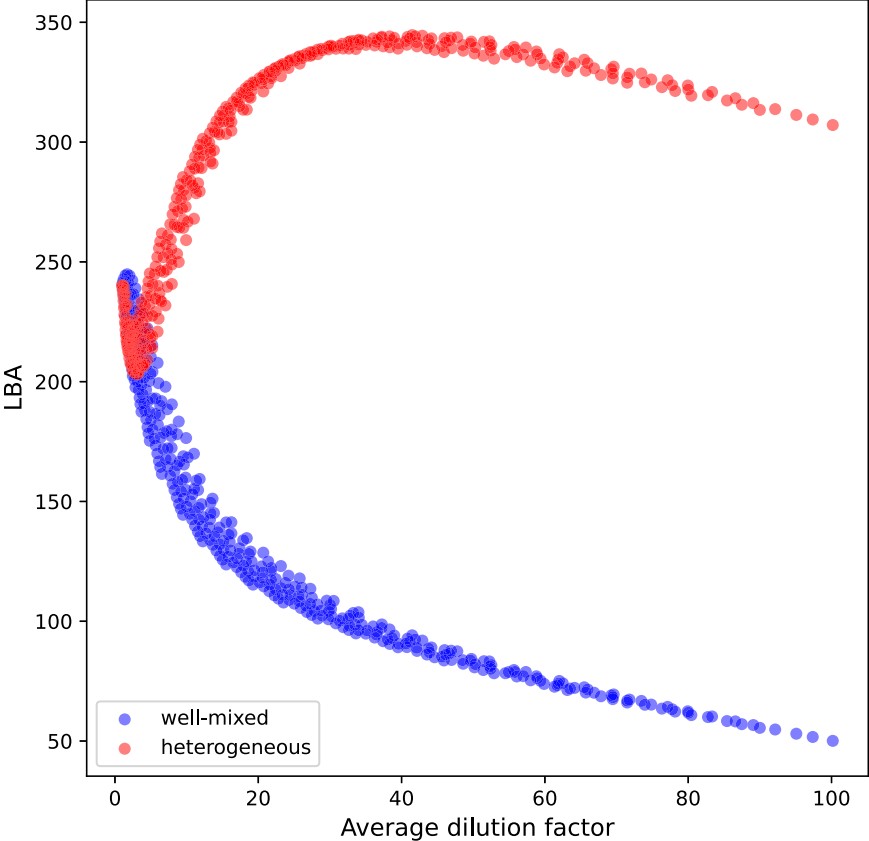

**Fig. 7 | Lagrangian biomass anomaly versus average dilution.** We show the Lagrangian biomass anomaly (LBA) measured in tonnes of carbon [Mg C] versus average dilution factor for ensembles of heterogeneous (red) and well-mixed (blue) patches for one month of integration. Each dot corresponds to a single simulation. For the well-mixed ensemble, the LBA decreases monotonically with dilution. In the heterogeneous ensemble the LBA presents a sharp and shallow minimum at low dilutions before a steady increase until reaching its maximum at intermediate dilutions. In the higher range of dilutions, the LBA of the heterogeneous ensemble is up to 600% larger than in the well-mixed one.

(Supplementary Fig. 6). We conclude that there is a dilution threshold that has to be passed to generate a level of spatial heterogeneity sufficient to abruptly enhance growth with respect to the well-mixed scenario. Above such threshold, the detrimental effect of the smearing of resource concentrations is overcompensated by spatial heterogeneity, allowing the LBA to rise for increasing dilution values.

The key point to understand the enhancement of LBA driven by spatial heterogeneity is the positive contribution of spatial variances and covariance to the consumer growth rate[27,28]. In general, Eqs. (32)-(36) highlight how the non-linear contribution of second moments can affect the derivatives of the means that can thus deviate importantly from the ones estimated using only first moments. In particular, as already shown for the SOIREE simulation (Fig. 5), the role of a positive spatial covariance seems to be crucial in enhancing biomass growth. We first note that, in a fertilized and growing patch, covariance is mostly positive due to fact that the water inside the patch is rich in both resource and concentration while the recently entrained water presents low concentrations of the two tracers. This configuration results in a positive spatial correlation between resource and biomass - i.e. positive covariance. Then, considering a simplified analytical model, it can be shown that the biomass growth rate, when calculated including spatial heterogeneity with positive covariance, is higher than the growth rate calculated only with the mean biomass and resource (Methods). This finally provides an heuristic explanation of why a positive covariance generated by dilution can increase the growth of the consumer.

Another aspect to consider when interpreting the LBA patterns is that dilution also increases the total patch volume. Indeed, large patches that underwent strong dilution, even if presenting a low biomass concentration, can attain larger LBA values relative to small patches with higher mean biomass concentrations (see Eq. (11)). This underlines the importance of a Lagrangian perspective to avoid misleading interpretations based only on Eulerian concentration fields i.e. focusing only on mean values without considering the volume associated with them.

As a confirmation of the robustness of our results, we find that the two distinct qualitative patterns of Fig. 6a versus Fig. 6b, i.e. a monotonous decrease versus the existence of a maximum ridge, are conserved in additional series of ensembles when varying the patch size, integration time, the parameters $\mu$ and $k$ and when considering a perfect recycling of resource by setting $\alpha = 1$ (Supplementary Figs. 3–5). For these simulations, when necessary, initial tracers concentrations are also changed consistently to ensure a steady-state surrounding. In the case where $\mu$ and $k$ are altered, the optimal LBA occurs at different strain/diffusion (Supplementary Fig. 4): this dependence is consistent with the hypothesis that different organisms can be better adapted to different degrees of turbulence[12,16,55,56]. Regarding the recycled fraction of the nutrients, we find that it can play a relevant role in the LBA budget especially in the late period of the bloom when the initial resource pulse is already depleted. Moreover, in an ensemble where we assume the extreme case of a "desert" surrounding - that is putting to zero all surrounding tracer concentration - we observe the same contrasting patterns between the heterogeneous and well-mixed ensembles (Supplementary Fig. 7). Finally, we also produce an ensemble using a quadratic mortality rate $m'$ in Eq. (10), substituting $mp_b$ with $m'p_b^2$. This allows us to implicitly account for some level of grazing on the

planktonic consumer[44,45]. Again, the qualitative difference between well-mixed and heterogeneous ensembles remains (Supplementary Fig. 8). We also note that, for this configuration, the well-mixed ensemble presents a tiny LBA peak at low but not null dilution rates, breaking the typical monotonous decrease observed in all the other model setups. This is consistent with a positive effect of dilution on phytoplankton growth observed in models that explicitly consider grazing dynamics, even in well-mixed conditions[16].

## Discussion

Previous Eulerian-based research has demonstrated that spatial heterogeneity can increase productivity relative to a well-mixed environment[25–28,40]. However, this has been only studied from the perspective of biological interactions and not of the drivers that create and modulate patchiness. On the other hand, Lagrangian approaches have been shown to be the most effective way to model and observe the mechanisms driving local bio-physical dynamics in the ocean since they focus on the real 'landscape' where a drifting ecosystem is evolving[11–14,17,19,57,58]. Our work establishes a connection between the study of ecological heterogeneity and the Lagrangian perspective of fluid flows and provides a theory to describe plankton ecosystems in the ocean.

The passage of weather systems, dust deposition events, and internal (sub)meso-scale physical processes continuously stimulate changes in the resource environment throughout the oceans. Our model reveals that such localized, transient enhancement of resources can lead to very different subsequent signatures in biomass depending upon the local strain, diffusion and surrounding tracer concentrations. Consequently, the measurable response (e.g. Chlorophyll concentration) of two resource injections of similar magnitude can be very different depending on the dilution rate. Thus, the relationships between remotely sensed Chlorophyll and produced biomass may be more complex than first intuition suggests. Nevertheless, it may be possible to account for aspects of this influence by interpreting the nature of the bio-physical environment.

Dilution has been already proposed in the past as a positive factor for plankton growth due to its effects of supplying nutrients or removing grazers[9,15,16,18]. Consistently, our model is able to reproduce such dynamics, in particular emulating the decrease of grazers pressure using a quadratic mortality (see Supplementary Fig. 8). However, here we show that dilution can also enhance biomass growth through only the physical mechanism of creating heterogeneity, without invoking other biologically driven mechanisms. We also foresee that, due to an increase of trophic efficiency caused by spatial heterogeneity[27,28], the Lagrangian biomass anomaly increment can be transferred to higher trophic levels (e.g., grazers). A more diluted and thus heterogeneous ecosystem would also be expected to have a reinforced stability that would ultimately boost the level of biodiversity that it can sustain[27,28,40,50]. From a community ecology perspective, entrainment can be quantitatively related to the rate at which organisms from outside the community disperse towards it[46]. This brings a key input - that is the dispersal rate - to community assembly theories allowing predictions of macro-ecological features such as diversity, Species-Abundance Distributions (SADs), Species-Area Relationships (SARs) and Taylor's law[24,46,47,49,50].

Our theoretical approach provides a bottom-up general framework to assess plankton ecology in the ocean from first principles. Indeed, a Lagrangian ecosystem can be regarded as the fundamental building block of more complex assemblages. Here, as proof of concept, we showed that our model can reproduce the features of the artificially fertilized bloom SOIREE[29]. However, our model can be applied to any Lagrangian ecosystem such as, for instance, the one illustrated in Fig. 1. Vertical dynamics can be included to describe exchanges across different depths[56]. In situations where strain and isotropic diffusion would not dominate the physical dynamics, higher moments of the deformation tensor could be taken into account to extend our model[59]. Our

framework could also be compared with alternative formulations in which an effective diffusivity is used to introduce characteristic time-scales in the system[57,58]. Also the complexity of the biochemical interactions can be escalated adding more tracers and new trophic layers[44,45], possibly including active plankton behavior. Moreover, instead of assuming 'mean-field' surrounding distributions, implementing multi-patches simulations would allow us to model how Lagrangian ecosystems interact with one another through the exchange of tracers while mixing and diluting. Though here we focused on a particular spatio-temporal scale, our approach can be adopted across wide ranges of physical and biochemical scales. This would permit us to explore how much a plankton ecosystem conserve the memory of its Lagrangian past unveiling its 'lifetime' i.e. for how long it can be considered significantly different from the surrounding[14]. More generally, this could ultimately help in revealing the effective spatio-temporal dynamics of an ecological perturbation across the seascape[22,57,58].

In summary, we present a framework that addresses the role of dilution and spatial heterogeneity (i.e. patchiness) on the response of plankton biomass to a local resource pulse. Nutrient injections are ubiquitous in the oceans and the interpretation of their biomass signatures contributes to our evaluations of ocean productivity. Perhaps unintuitively, we find that lateral dilution of such a feature can significantly enhance the integrated biomass anomaly due to the local generation of patchiness. These results therefore offer a significant addition in our understanding of bloom dynamics and are crucial when considering natural or deliberate nutrient fertilization events. In particular, our study shows that neglecting patchiness leads to a several-fold underestimate of the integrated biomass response to a resource injection. Hence we believe that accounting for dilution and unresolved patchiness, by a synergic use of remote-sensing observations, drifter and dye releases, is an important goal for biogeochemical sampling strategies and future modeling approaches.

## Methods

### Geometric description of a Lagrangian patch

We describe a Lagrangian patch as a two-dimensional evolving ellipse encompassing the majority of its surface[10,11,21,30,34]. To this aim, we associate to the patch a concentration of an ideal passive tracer that covers it at the initial time. This corresponds to theoretically dying all the molecules of the patch at a given time and tracking the evolution of the dye concentration at subsequent times. We denote such dye-patch concentration by $\theta(\mathbf{x}, t)$ and we assume that the isolines of such distribution describe elliptic areas. We define $L(t)$ and $W(t)$ as the spatial variances of the distribution $\theta(\mathbf{x}, t)$ along the axes of the ellipse[10,21,34]. We thus identify $2W(t)$ as the patch width (the minor axis of the ellipse) and $2L(t)$ as the the patch length (the major axis of the ellipse) while the patch center of mass is denoted as $\mathbf{X}(t)$ (see Fig. 2). The characteristic patch size is defined as: $S(t) \equiv W(t) + L(t)$. Consequently, the area of the ocean surface associated with the patch is:

$$A(t) = \pi W(t) L(t) \tag{12}$$

### Lagrangian advection-diffusion equation and patch physical evolution

We take a Lagrangian perspective focusing on the trajectory and the modification of the water patch. To this aim we chose a reference frame that is translating and rotating with the patch. In this way all the rigid-like movements - that are the ones that do not change the relative positions of the fluid elements in the patch - are ignored[32]. Since we consider here incompressible flows, we set the divergence to zero and the velocity field can be locally associated to an elliptically symmetrical stagnation flow. The associated stirring effect on the patch, at the spatial scale $S(t)$, can be described by a strain rate coefficient $\gamma(\mathbf{X}(t), S(t), t) \equiv \gamma(t)$ (see Fig. 3). Advection, rotation and stirring

are not responsible for the dilution of the patch in the surrounding since they are not directly related to mixing. Indeed under their action, the area associated to the patch remains constant in time. On top of such deterministic stirring dynamics we superimpose the effect of diffusion related to unresolved scales of the velocity field smaller of the typical patch size $S(t)$. We denote the size dependent diffusion as $\kappa((\mathbf{X}(t), S(t), t) \equiv \kappa(t)$ (see Fig. 3).

We derive then the advection-diffusion equation for the distribution $\theta(\mathbf{x}, t)$[10,21,30,34]:

$$\frac{\partial \theta(\mathbf{x}, t)}{\partial t} + \mathbf{v}(\mathbf{x}, t) \cdot \nabla \theta(\mathbf{x}, t) = \kappa(t) \nabla^2 \theta(\mathbf{x}, t), \qquad (13)$$

where $\mathbf{v}(\mathbf{x}, t)$ is the velocity. In a proper Lagrangian frame of refs. 21,32 when the contracting direction is aligned with the x-axis and the expanding one with the y-axis, Eq. (13) becomes:

$$\frac{\partial \theta}{\partial t} - \gamma(t) x \frac{\partial \theta}{\partial x} + \gamma(t) y \frac{\partial \theta}{\partial y} = \kappa(t) \nabla^2 \theta, \qquad (14)$$

where, for brevity, we omitted in the notation the temporal and spatial dependence of $\theta(\mathbf{x}, t)$.

The zeroth and second order spatial integrals of the tracer distribution $\theta$ are:

$$M_0^x(t) = \int \theta(\mathbf{x}, t) \Big|_{y=0} dx \quad ; \quad M_2^x(t) = \int x^2 \theta(\mathbf{x}, t) \Big|_{y=0} dx, \qquad (15)$$

$$M_0^y(t) = \int \theta(\mathbf{x}, t) \Big|_{x=0} dy \quad ; \quad M_2^y(t) = \int y^2 \theta(\mathbf{x}, t) \Big|_{x=0} dy. \qquad (16)$$

Hence, the squares of the width and length of the patch can be expressed as:

$$W^2(t) = \frac{M_2^x(t)}{M_0^x(t)}, \qquad (17)$$

$$L^2(t) = \frac{M_2^y(t)}{M_0^y(t)}. \qquad (18)$$

Deriving in time the above expressions and integrating in space Eq. (14), we obtain the time evolution for the patch width and length[11,21,34]:

$$\frac{\partial W^2(t)}{\partial t} = +2\kappa(t) - 2\gamma(t) W^2(t), \qquad (19)$$

$$\frac{\partial L^2(t)}{\partial t} = +2\kappa(t) + 2\gamma(t) L^2(t). \qquad (20)$$

Combining the above equations with Eq. (12) we finally obtain the increase rate for the patch area:

$$\frac{dA(t)}{dt} = \pi \kappa(t) \left[ \frac{W^2(t) + L^2(t)}{W(t) L(t)} \right]. \qquad (21)$$

## Entrainment effects on tracer distributions
Diffusion at the patch boundaries causes entrainment of surrounding waters in the patch. The rate at which this process happens can be estimated from the rate at which the patch area is growing i.e. from $dA(t)/dt$. We derive here the contribution of such processes on the evolution of first and second moments of tracers inside the patch. In this section the patch is explicitly indicated with `pat` while its surrounding is indicated with `sur`. For an interval of time $\Delta t$ the area of the patch at time $t$ will increase from $A$ to $A + \Delta A$. For mass conservation, the surface intruded in the patch bringing waters with different composition, should correspond exactly to $\Delta A$. In the following we derive the equations describing how means, variances and covariances change when we merge the two regions `pat` and $\Delta$`pat`, of surface $A$ and $\Delta A$ respectively, with different tracer compositions (see Fig. 4 and Supplementary Fig. 1).

Let's derive the equation for the mean values first. By definition we can write:

$$\langle p_i(\mathbf{x}, t) \rangle_{\text{pat}} = \frac{1}{A} \int_{\text{pat}} p_i(\mathbf{x}, t) ds, \qquad (22)$$

$$\langle p_i(\mathbf{x}, t) \rangle_{\Delta\text{pat}} = \frac{1}{\Delta A} \int_{\Delta\text{pat}} s_i(\mathbf{x}, t) ds, \qquad (23)$$

where in the second equation the integrand is $s_i(\mathbf{x}, t)$ because $\Delta$`pat` is intruding from the surrounding of the patch. Considering the mean value of both areas merged at time $t + \Delta t$ we have:

$$\langle p_i(\mathbf{x}, t + \Delta t) \rangle_{\text{pat} \cup \Delta\text{pat}} = \frac{1}{A + \Delta A} \left( A \langle p_i(\mathbf{x}, t) \rangle_{\text{pat}} + \Delta A \langle s_i(\mathbf{x}, t) \rangle_{\text{sur}} \right). \qquad (24)$$

Using the definition of derivative $\frac{df(t)}{dt} = \frac{f(t + dt) - f(t)}{dt}$, taking the limits $\Delta A \to dA \to 0$ and $\Delta t \to dt \to 0$, we obtain:

$$\frac{d \langle p_i(\mathbf{x}, t) \rangle_{\text{pat}}}{dt} = \frac{1}{A(t)} \left( \frac{dA(t)}{dt} \right) \left( \langle s_i(\mathbf{x}, t) \rangle_{\text{sur}} - \langle p_i(\mathbf{x}, t) \rangle_{\text{pat}} \right) \qquad (25)$$

With a similar approach and using the definition of spatial variance we can derive the equation for the derivative of the variances. The variance of both volumes merged at time $t + \Delta t$ is:

$$\langle p_i'(\mathbf{x}, t + \Delta t)^2 \rangle_{\text{pat} \cup \Delta\text{pat}} = \frac{1}{A + \Delta A} \left( \int_{\text{pat}} p_i(\mathbf{x}, t)^2 ds + \int_{\Delta\text{pat}} s_i(\mathbf{x}, t)^2 ds \right) - \left( \langle p_i(\mathbf{x}, t + \Delta t) \rangle_{\text{pat} \cup \Delta\text{pat}} \right)^2 \qquad (26)$$

Developing the integral terms and using Eq. (24):

$$\langle p_i'(\mathbf{x}, t + \Delta t)^2 \rangle_{\text{pat} \cup \Delta\text{pat}} = \left( \frac{1}{A + \Delta A} \right) \left( A \left( \langle p_i'(\mathbf{x}, t)^2 \rangle_{\text{pat}} + \langle p_i(\mathbf{x}, t) \rangle_{\text{pat}} \right) + \Delta A \left( \langle s_i'(\mathbf{x}, t)^2 \rangle_{\text{sur}} + \langle s_i(\mathbf{x}, t) \rangle_{\text{sur}} \right) \right) - \left( \frac{1}{A + \Delta A} \right)^2 \left( A \langle p_i(\mathbf{x}, t) \rangle_{\text{pat}} + \Delta A \langle s_i(\mathbf{x}, t) \rangle_{\text{sur}} \right)^2. \qquad (27)$$

Developing all terms, taking the limits $\Delta A \to dA \to 0$ and $\Delta t \to dt \to 0$ and using the definition of derivative we obtain:

$$\frac{d \langle p_i'(\mathbf{x}, t)^2 \rangle_{\text{pat}}}{dt} = \frac{1}{A(t)} \left( \frac{dA(t)}{dt} \right) \left( \langle s_i'(\mathbf{x}, t)^2 \rangle_{\text{sur}} - \langle p_i'(\mathbf{x}, t)^2 \rangle_{\text{pat}} + \left( \langle s_i(\mathbf{x}, t) \rangle_{\text{sur}} - \langle p_i(\mathbf{x}, t) \rangle_{\text{pat}} \right)^2 \right). \qquad (28)$$

Finally, generalizing Eq. (28), we have an expression for the derivative of the covariance between tracer $i$ and $j$:

$$\frac{d \langle p_i'(\mathbf{x}, t) p_j'(\mathbf{x}, t) \rangle_{\text{pat}}}{dt} = \frac{1}{A(t)} \left( \frac{dA(t)}{dt} \right) \left( \langle s_i'(\mathbf{x}, t) s_j'(\mathbf{x}, t) \rangle_{\text{sur}} - \langle p_i'(\mathbf{x}, t) p_j'(\mathbf{x}, t) \rangle_{\text{pat}} + \left( \langle s_i(\mathbf{x}, t) \rangle_{\text{sur}} - \langle p_i(\mathbf{x}, t) \rangle_{\text{pat}} \right) \left( \langle s_j(\mathbf{x}, t) \rangle_{\text{sur}} - \langle p_j(\mathbf{x}, t) \rangle_{\text{pat}} \right) \right). \qquad (29)$$

## Internal mixing within the patch

A passive tracer in a turbulent flow is subjected to mixing processes that tend to homogenize its concentration in time. Several approaches have been developed to theoretical model the decay of the moments of a tracer distribution[35,41–43]. From a patch perspective, internal mixing does not affect spatial means but it contributes to smooth variances and covariances. In particular, the decay rate of tracer second moments can be related to the diffusion acting at the corresponding spatial scale[41,43]. Using that the diffusion coefficient $\kappa(t)$ represents the effective diffusion at the scale of the patch size $S(t)$, we conclude that the decay rate of tracers variances and covariances in the patch is[35,42]:

$$\langle p_i'(\mathbf{x},t)p_j'(\mathbf{x},t)\rangle \sim e^{-\frac{\kappa(t)}{S(t)^2}t}. \tag{30}$$

From the above functional dependence we finally derive the expression for the internal mixing contribution to the time derivative of second moments:

$$\frac{d\langle p_i'(\mathbf{x},t)p_j'(\mathbf{x},t)\rangle}{dt} = -\frac{\kappa(t)}{S(t)^2}\langle p_i'(\mathbf{x},t)p_j'(\mathbf{x},t)\rangle. \tag{31}$$

## First and second moments contributions to biological dynamics

Based on the Reynold's decomposition for tracer distributions of Eq. (4) we can derive the contribution of spatial means, variances and covariances to Eqs. (9) and (10). To this aim, we use a closure method to provide analytical expressions for time derivatives of first and second moments in the patch[25–28,40]. In the following, to simplify notation, we omit the dependence of tracer distributions on $t$ and $\mathbf{x}$.

The equations for the evolution of the means are:

$$\frac{d\langle p_r\rangle}{dt} = -\nu\frac{\langle p_r\rangle\langle p_b\rangle}{(\langle p_r\rangle+k)} + \nu k\frac{\langle p_b\rangle\langle p_r'^2\rangle}{(\langle p_r\rangle+k)^3} - \nu k\frac{\langle p_r'p_b'\rangle}{(\langle p_r\rangle+k)^2} + \alpha m\langle p_b\rangle, \tag{32}$$

$$\frac{d\langle p_b\rangle}{dt} = +\nu\frac{\langle p_r\rangle\langle p_b\rangle}{(\langle p_r\rangle+k)} - \nu k\frac{\langle p_b\rangle\langle p_r'^2\rangle}{(\langle p_r\rangle+k)^3} + \nu k\frac{\langle p_r'p_b'\rangle}{(\langle p_r\rangle+k)^2} - m\langle p_b\rangle. \tag{33}$$

The evolution of the variances are:

$$\frac{d\langle p_r'^2\rangle}{dt} = -2\nu k\frac{\langle p_b\rangle\langle p_r'^2\rangle}{(\langle p_r\rangle+k)^2} - 2\nu\frac{\langle p_r\rangle\langle p_r'p_b'\rangle}{(\langle p_r\rangle+k)} + 2\alpha m\langle p_r'p_b'\rangle, \tag{34}$$

$$\frac{d\langle p_b'^2\rangle}{dt} = +2\nu\frac{\langle p_r\rangle\langle p_b'^2\rangle}{(\langle p_r\rangle+k)} + 2\nu k\frac{\langle p_b\rangle\langle p_r'p_b'\rangle}{(\langle p_r\rangle+k)^2} - 2m\langle p_b'^2\rangle. \tag{35}$$

Similarly, we can obtain the evolution of the covariance:

$$\frac{d\langle p_r'p_b'\rangle}{dt} = \nu\frac{\langle p_r\rangle}{(\langle p_r\rangle+k)}\left(\langle p_r'p_b'\rangle - \langle p_b'^2\rangle\right)$$
$$+ \nu k\frac{\langle p_b\rangle}{(\langle p_r\rangle+k)^2}\left(\langle p_r'^2\rangle - \langle p_r'p_b'\rangle\right) + m\left(\alpha\langle p_b'^2\rangle - \langle p_r'p_b'\rangle\right). \tag{36}$$

## Bio-physical parameters setting for ensemble simulations

We detail below the setting of physical and biological parameters used for the main ensemble simulations (Figs. 6 and 7). Other ensemble simulations to address the model sensitivity using different sets of parameters are presented in the Supplementary Information (Supplementary Figs. 3, 4, 5, 7 and 8).

We set the Lagrangian ecosystem model to study the evolution of a horizontal circular patch of initial diameter of $S(0) = 10$ km and constant thickness of 10 m. We track its evolution over a time window

(i.e. the integration time) of $\tau = 30$ days with a time-step of ~ 14 min. The ranges of realistic values of initial strain and diffusion used are based on in-situ observations[11,30,36,38,39]. They corresponds to: $0.01 < \gamma < 0.6$ day$^{-1}$ and $0.01 < \kappa < 0.6$ km$^2$day$^{-1}$, respectively. We then implement specific scaling laws of $\gamma$ and $\delta$ for the spatial scales of 10–100 km spanned by our ensemble simulations:

$$\gamma(t) = f_\gamma[S(t)] = \alpha S(t)^{-\frac{2}{3}}, \tag{37}$$

$$\kappa(t) = f_\kappa[S(t)] = \beta S(t), \tag{38}$$

where $\alpha$ and $\beta$ are chosen in a way that $\gamma(t=0)$ and $\kappa(t=0)$ match realistic values at the scale of the initial patch size $S(t=0)$. For the study of the SOIREE experiment we use as initial value of strain and diffusion $\gamma = 0.12$ and $\kappa = 0.1$, respectively.

We identify the resource $p_r$ with iron and the consumer $p_b$ with phytoplankton. We also assume that resource and consumer can be reasonably approximated to be passive tracers with no active behavior. We do not model resource recycling ($\alpha = 0$) with exception of the sensitivity analysis reported in Supplementary Fig. 5 in which we instead use a complete remineralization rate ($\alpha = 1$). The Fe:C ratio used is $10^{-5}$[15,44]. The initial iron concentration in the patch is $1\,\mu$mol/m$^3$ and $0.1\,\mu$mol/m$^3$ at the surrounding while the initial phytoplankton concentration in iron currency, both in the patch and at the surrounding, is $0.0249\,\mu$mol/m$^3$[13,29]. Initial variances and covariance are set to zero. The maximum phytoplankton growth rate and its linear mortality rate are: $\nu = 1.05$ day$^{-1}$ and m = 0.05 day$^{-1}$[15,29,44,52]. The half-saturation constant for iron is: $k = 2\,\mu$mol/m$^3$[53,54].

## Simplified analytical model of an heterogeneous patch

Here we introduce a simplified model to investigate the role of positive covariance for biomass growth. Let's consider an analytical model of a patch composed by just two sub-regions of equal size. In sub-region 1 the concentrations of resource is $r_1$ and of biomass is $b_1$, respectively we have $r_2$ and $b_2$ for sub-region 2. If we identify the total growth rate of the patch as the average of the growth rates of the two sub-regions we would have:

$$\frac{\nu}{2}\left[\frac{r_1 b_1}{r_1+k} + \frac{r_2 b_2}{r_2+k}\right] \tag{39}$$

If we instead do the opposite, i.e., average first the concentrations of the two sub-regions and only after compute a single growth rate for the entire patch, we have:

$$\frac{\nu}{4}\left[\frac{(r_1+r_2)(b_1+b_2)}{\frac{r_1+r_2}{2}+k}\right] \tag{40}$$

Then, we can consider the case in which we have a positive spatial covariance in the patch by setting:

$$r_1 = r + \delta r \quad ; \quad b_1 = b + \delta b \tag{41}$$

$$r_2 = r - \delta r \quad ; \quad b_2 = b - \delta b \tag{42}$$

The difference of the two different growth rate above, i.e. expression (39) - expression (40), becomes:

$$\Delta = \nu\frac{(k+r)\delta b - b\delta r}{(k+r)(k+r-\delta r)(\delta r+k+r)}k\delta r \tag{43}$$

Assuming that $k + r > \delta r$, then $\Delta$ is positive if and only if $(k+r)\delta b > b\delta r$.

**Reporting summary**

Further information on research design is available in the Nature Portfolio Reporting Summary linked to this article.

## Data availability

All external data used, such as observations or laboratory measures, are freely available from the corresponding publications cited along the text.

## Code availability

The code used to perform all the simulations presented in the manuscript is freely available on-line at: https://github.com/serjaaa/lagrangian-ecosystem-model.

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

## Acknowledgements

E.S.-G. thanks Claudia Payrato-Borras for providing feedback on the initial conceptualization of the ecosystem model. E.S.-G., S.D and M.J.F. are very grateful for support from the Simons Foundation: the Simons Collaboration on Computational BIOgeochemical modeling of Marine EcosystemS (CBIOMES #549931 to M.J.F.). RMG acknowledges support from FAPESP through a BIOTA Young Investigator Research Grant no. 2019/05523-8 and ICTP-SAIFR grant no. 2016/01343-7; Instituto Serra-pilheira through grant Serra-1911-31200, and the Simons Foundation through grant no. 284558FY19. This work was partially funded by the Center of Advanced Systems Understanding (CASUS), which is financed by Germany's Federal Ministry of Education and Research (BMBF) and by the Saxon Ministry for Science, Culture and Tourism (SMWK) with tax funds on the basis of the budget approved by the Saxon State Parliament.

## Author contributions

E.S.-G. developed the theory and performed numerical simulations. E.S.-G., R.M.G., S.D., and M.J.F. interpreted the results and wrote the manuscript.

## Competing interests

The authors declare no competing interests.
