## [Peer Review file · Nature Communications]

REVIEWER COMMENTS

Reviewer #1 (Remarks to the Author):

Review of "A Lagrangian model for drifting ecosystems reveals heterogeneity-driven enhancement of marine plankton blooms" by Ser-Giacomi et al

In this manuscript, the authors propose a new framework ('formalism?') to account for subscale-heterogeneity in Lagrangian simulations, by considering patches of firing size in which within-patch variation can be non-zero. They show the potential of this framework by applying it to an iron fertilization case.

This is a very exciting manuscript, that I have read with much interest! I think that it will make a huge impact in the Lagrangian Oceanography community, especially those who aim to marry ocean physics and biogeochemistry. With the start of the EU-funded NECCTON project, this manuscript comes at exactly the right time to support big advances in Lagrangian biogeochemical oceanography. I thus strongly support the publication of this manuscript, after my considerations below have been taken into account

I have five major points that I would like the authors to consider

1. One of the key assumptions underpinning the framework is that plankton and nutrients drift passively with water masses. While that may be true for nutrients, it's not per se the case for plankton. How would the framework change for active plankton? Or is that just a limitation?
2. I don't understand why the method is fully described in 2D in section II.A. How would this extend to 3D?
3. The phrase 'water patch' doesn't get a proper definition until much later in the manuscript. To not confuse readers, I suggest here already defining what a water patch is. And what happens in the limit of S going to zero? Does the framework then still hold?
4. The authors focus exclusively on the first and second moment of the patch properties, which begs the question why no attention is given to higher-order moments? What is the rationale behind that?
5. The 3D plots in Fig 6 and the Supplementary Material are really difficult to comprehend and appreciate in detail. Given that there is a direct match between colour and z-axis, why not show as 2D heat-maps (i.e. remove the z-axis and only show colour)?

Furthermore, I have some minor points listed below:

- Abstract: Quantify 'several fold'?
- Abstract: who is 'we' in the second-last line?
- line 9: 'important' in which sense?
- line 19: how is 'grazer' defined? That concept is not introduced before
- first line below line 90 (note no line-numbers here...): 'patch dilution is also not properly defined here
- line 101: why would temperature not explicitly depend on patch evolution? If the patch becomes very large, then internal temperature gradients become important? Or is this again a matter of the definition of a water patch?
- line 107: why do the authors only assume linear superposition of the terms? Why no cross terms? Would that even be possible, in this framework?
- line 134: make explicit here that p_r is iron, and p_b is phytoplankton in this case?
- Table 1: Why is this table important enough for the main manuscript? And why are the units not in S.I.?
- line 136: gamma and kappa miss a unit here
- line 166: mention here already what happens if s_b is not uniform?
- line 195: mention here that Eq 34 is in the Methods section?
- line 195: "we finds that _this_ quantity"?

- line 197: why does this dilution threshold exist? Is there an intuitive reason?
- line 207: should 'concentration' here be 'biomass'?
- Fig 7: The axes have no units
- line 278: is 'formalism' the best word here? Why not call it a 'framework'?
- line 281: use 'counterintuitively', instead of 'unintuitively'?
- line 289: what is meant with 'majority of its surface'? That is a very vague statement
- Eqs 40 and 41: These are not equations in the sense that there is no left-hand term

Reviewer #2 (Remarks to the Author):

The manuscript by Ser-Giacomi et al. presents a new theoretical framework for the description of the dynamics of isolated plankton patches. Specifically, it presents a set of equations which allow reconstructing the evolution of the patch global properties (mean and variance of the biomass) while quite explicitly taking into account the patch internal variability as well as the exchanges with the environment in background.

The manuscript thus addresses the important question of interpreting both natural and artificial fertilization events and to predict their fates in terms of responses to both the background flow and the specific ecological constraints. It is an important matter for many reasons. The most important it is that in ecology we miss a theoretical framework that describes the "seascape" with its specificities and this study contributes to its construction, indeed. Further, given the renewed interest in exploiting the oceans for reducing the atmospheric CO₂ (carbon dioxide removal via, eg, artificial upwellings), it is urgent to build the knowledge that would allow predicting the outcome of such exercises.

The manuscript is fairly well written but it still needs some work, especially in terms of clarifying the assumptions behind the conceptual framework. Moreover, the discussion of the results could profit of an improved discussion of past studies which already presented a sensitivity of the mean values to diffusion, eg, by emphasizing the relevant time scales, as for instance, in Pasquero, GRL, doi:10.1029/2005GL023662 (2015) (and papers cited in it).

Finally, the theory is fully developed for the case of a relatively simple flow. It is therefore necessary to discuss its limitations when applying it to real cases. What are the temporal and spatial scales for which it is supposed to be valid? How sensitive is the evolution of biomass to errors/approximations in estimating the strain rates? etc.?

Specific comments:

Abstract:

The oxygen budget of the oceans is nearly zero. Better not to mention it here and in the following.

I do not fully understand what is a "Lagrangian patch". In fact, a patch of water is always transported by currents. I would thus avoid adding "Lagrangian" (also later in the text). The Lagrangian one is a theoretical approach and not a physical property, I would say.

"the model resolves": I would call it the theory and not the model to avoid generating confusion.

"Propagation" is too vague. Evolution? Change in time?

Main text:

L4 Why However?

L23 Evolution intended as change in time. Better to be explicit.

L40 Patchiness is here intended as generic heterogeneity. I would use the latter.

L39 and following: This study is the first that proposes an integrated approach to the effect of diffusion and strain on a plankton patch but it is not the first study that deals with this matter (most of them cited later on in the main text). The point here is the quantification of the impact of heterogeneities on the mean properties.

In addition, a better justification is needed about its focus on the mean biomass and not on the integrated biomass evolution, the latter being ecologically more relevant, possibly. Indeed, the "observable" is the local biomass concentration and, thus, I agree it should be focus. (In addition, combining the estimate of the mean value with the area of the patch the total biomass is readily obtained, as in the LBA formulations presented later on in the manuscript).

Everywhere in the manuscript: use Ellipse instead of Ellipsis.

L54 Is $S = L + W$ or $(L+W)/2$?

L45 and following (including the associated Methods): My understanding is that the theoretical framework is built by first describing the patch as made of a passive tracer (and not a water concentration, as mentioned in the Methods) Theta whose isolines are ellipses, as done in the past for interpreting tracer release experiments. Building upon the equation for the passive tracer evolution an equation is derived to describe the evolution of the patch area (defined using the moments of Theta) as a function of the diffusivity.

The latter equation is then used to integrate the equations of the active tracer (the plankton). It is a heuristic approach, evidently, since an active tracer may diffuse differently than a passive one. Therefore, the text describing it has to be clarified since it is currently difficult to grasp the main "conceptual" framework.

As illustrated in Fig. 3, the diffusion increases the patch. But what is actually "diffusing" is Theta. It is somewhat confusing.

L63 (eq (1)): What is the sensitivity of DA/dt to changes in k and γ , respectively? Why the role of k is explicit, the contribution of γ is actually hidden in the eqs for the evolution of W and L and it is difficult to grasp who is actually regulating the "dilution".

L66 Can we really state that it is proportional to the perimeter while not having a closed equation of the perimeter value?

L103 and following: Eq. 8 is a bit misleading and also obvious. The novelty here is made of the full equations for mean and variance (which are then integrated numerically) and thus it is necessary to have those equations presented here instead of the equation of plankton local concentrations.

Table 1: It would be better to have values in m instead of km.

L133 and following: it would be better to present at least one figure (Chl from satellite?) showing the choice of the estimate of the L and W for the real case. There

L143-144: Please, define dilution.

L150-151: Is it really surprising to find a relation between heterogeneity and biomass growth, given the non-linear nature of the growth term? In fact, this is the global impact of the effect of "patchiness" studied in the past by Martin, Neufeld etc and it should be discussed here or it should be mentioned that it is further discussed in Section C.

Moreover, is this flow field able to generate patchiness? I would say that heterogeneity is a better term here.

L152: I understand that the dynamical coupling is considered as "implicit" but you could reach a deeper understanding of the system by discussing the variation in time of the values of each term (eg, mean growth, etc) of the equation that describes the "mean" value () time evolution, in principle. This exercise could inform also the interpretation of the various sensitivity studies.

L161: the analysis of the LBA is interesting but it is not an estimate of the total biomass produced. And as demonstrated in the fig. 10, it depends upon the interval of integration (τ ; not explicit in fig. 6 and 7, by the way). To fully understand the observed sensitivity it may be useful analyzing also the max of A^* (

- $\langle s \rangle$).

L236 and following: The Discussion emphasizes the effect of dilution. It would profit a more in depth discussion of the literature (eg, Pasquero's papers mentioned above).

It should also include a discussion about the mechanisms that are known to generate "patchiness" in the oceans, such as filamentation and more generally submesoscales processes. These processes have well defined time and spatial scales and I suspect those scales set specific constraints on the applicability of the approach proposed here. More broadly, a discussion the various mechanisms acting at various scales in the oceans should be briefly discussed here to let the reader understand in which cases this approach is actually useful (eg, on scales of km or hundreds of km?).

L261: the use of the term "migrate" (coming from neutral ecological theories) is a bit misleading. I quite understand the argument here but it should be clarified that it is intended as transport into the system.

L321 and following: It is an exact derivation but the entire outcome depends upon the definition of A and its evolution in time. In practice, how would A be determined? As an isoline of Chl concentration from satellite data? Or as the evolution of the area based upon the strain (eg, from altimetry) and diffusion rates?

L364 The diffusion of an active tracer does not follow specific laws since the tracer reacts while diffusing (eg, Pasquero 2015). Please discuss here the assumptions made in eq. 39.

Eq. 28. Is the second term on rhs $\langle p_i \rangle_{pat}$ or $\langle p_i \rangle_{pat}^2$?

Fig. 9 Right panel. The sensitivity analysis is very qualitative and it may be improved by adding a RMS estimate, I would like to suggest. And it shows lower biomass for a higher dilution thus suggesting that the real case is not representative of the biomass enhancement via dilution discussed in the main text.

POINT-BY-POINT RESPONSE TO REVIEWER 1

(Referee's comments are reported in *italic* while our answers immediately follow in **red font**.)

I have five major points that I would like the authors to consider

- 1. One of the key assumptions underpinning the framework is that plankton and nutrients drift passively with water masses. While that may be true for nutrients, it's not per se the case for plankton. How would the framework change for active plankton? Or is that just a limitation?*

We agree with the referee on the fact that plankton is not a completely passive tracer. However, almost 100% of biogeochemical and plankton models assume that phytoplankton and small zooplankton can be reasonably approximated to be passive. On the other hand, active plankton behavior could be in principle considered as a future extension of the model. For instance, vertical migration of zooplankton could be parametrized as a net flux across the patch boundaries. In the revised manuscript we state that we approximated plankton as a passive tracer, and we also mentioned the possibility of including non-passive dynamics (lines 285, 389).

- 2. I don't understand why the method is fully described in 2D in section II.A. How would this extend to 3D?*

Very good question, thanks. The present version of model is 2-dimensional (resolving only horizontal dynamics). Notably, on average, vertical dynamics is much slower than the horizontal one in the ocean (except, of course, very relevant exceptions) and this is the reason why we kept our approach 2-dimensional. This is also the approach followed by related studies such as (Abraham et al. 2000). However, we are currently working on a 3-dimensional extension of our framework to include vertical diffusion and advection. We believe that, due to the already complex theoretical developments presented in the manuscript, such extension will fit better in another publication, and we mentioned this in the discussion (line 280).

- 3. The phrase 'water patch' doesn't get a proper definition until much later in the manuscript. To not confuse readers, I suggest here already defining what a water patch is. And what happens in the limit of S going to zero? Does the framework then still hold?*

We added some text in section I and II.A to introduce the general concept of a water patch in the ocean (lines 14, 52, 55, fig. 2). Our initial approach is the same as the one presented by, for instance, Ledwell et al. 1998. A patch is initially defined as a circular portion of the ocean and the associated body of water is tracked in time using elliptical isolines. About the limit for S going to zero our answer is yes: indeed, all the quantities and equations scale with the logarithmic derivative of the area. This means that, if strain and diffusion are also changing in physically meaningful sense, Eq. (5 - 6) will not present divergences. In other words, all the dynamics modeled by our framework can be rescaled smoothly as a function of the size of the patch.

- 4. The authors focus exclusively on the first and second moment of the patch properties, which begs the question why no attention is given to higher-order moments? What is the rationale behind that?*

This is a very interesting topic indeed. Higher moments of the tracer distributions and its contribution to biochemical dynamics can be derived by a truncation of the Taylor expansion at higher orders i.e. by using a higher-order closure method. Moreover, some very interesting works by Okubo also studied higher-order moments in the Lagrangian framework from a purely physical perspective. Extending Okubo's approach based on a higher-moment closure approximation to incorporate biochemical reactions could reveal new and more complex biophysical dynamics. We are seriously considering this direction as a possible next step of this project. In this study we only focused on first and second moments since we consider that it is already a new perspective (almost all plankton models consider only the first moments) that deserves publication on its own. We added a sentence with referenced bibliography about this (line 281).

- 5. The 3D plots in Fig 6 and the Supplementary Material are really difficult to comprehend and appreciate in detail. Given that there is a direct match between colour and z-axis, why not show as 2D heat-maps (i.e. remove the z-axis and only show colour)?*

We thank the referee for pointing out this issue and we now agree that a more “classical” 2-d heatmap plot gives a clearer understanding of how LBA behaves. Following this suggestion, we changed Fig. 6 accordingly. We also added a third panel showing the difference, for each pixel, between the heterogeneous ensemble and the well-mixed one. However, we decided to leave the 3-d representation in the Supplementary material to highlight the feature of the ridge of maximum LBA (shape, size, steepness, and concavity). We think that combining these two visualizations provides a more complete and richer description of the dynamics.

Furthermore, I have some minor points listed below:

Thanks a lot for the detailed revision. We included/considered most of these suggestions and edited the manuscript accordingly. Below we answer to some of the comments that needed further explanations.

line 107: why do the authors only assume linear superposition of the terms? Why no cross terms? Would that even be possible, in this framework?

This is because we isolated processes that are independent (e.g. phytoplankton consuming nutrient is a process independent to patch physical strain). In this way cross-terms – i.e. interactions among them – are absent.

- Table 1: Why is this table important enough for the main manuscript? And why are the units not in S.I.?

The table can be useful to quickly compare our realistic model setting with data from different observations or simulations. The units used are common in physical oceanography.

- line 166: mention here already what happens if s_b is not uniform?

The LBA depends only on first moments. Thus, is not directly affected by the variance of the surrounding concentration. Therefore, we did not refer to it here. However, the variance of external waters, as the referee pointed out is fully considered in equations 6 and 7.

- line 197: why does this dilution threshold exist? Is there an intuitive reason?

This is indeed a very good question! We do not have a clear answer to that. Most likely non-linear feedbacks in the dynamics drives such a sharp transition but we did not manage to explicitly identify it.

- Fig 7: The axes have no units

We preferred to not add unit on the figure axes for graphical clarity, but we always specify them in the caption. Dilution has no units since it is adimensional.

POINT-BY-POINT RESPONSE TO REVIEWER 2

(Referee's comments are reported in *italic* while our answers immediately follow in **red font**.)

The manuscript is fairly well written but it still needs some work, especially in terms of clarifying the assumptions behind the conceptual framework. Moreover, the discussion of the results could profit of an improved discussion of past studies which already presented a sensitivity of the mean values to diffusion, eg, by emphasizing the relevant time scales, as for instance, in Pasquero, GRL, doi:10.1029/2005GL023662 (2015) (and papers cited in it).

As already stated in the cover letter, we agree that we missed to discuss some conceptual aspects of the theory presented. To solve this issue, we reformulated parts of the text and included the references suggested (lines 14, 52, 55, 306, fig. 2). About the physical scales covered by this study we stress that, even though we focused on mesoscale dynamics in our ensemble simulations, our approach is general and can be extended to any scale using different scaling functions for strain and diffusion. This can be done by modifying the functional forms of Eqs. (38-39).

Finally, the theory is fully developed for the case of a relatively simple flow. It is therefore necessary to discuss its limitations when applying it to real cases. What are the temporal and spatial scales for which it is supposed to be valid? How sensitive is the evolution of biomass to errors/approximations in estimating the strain rates? etc.?

We fully agree that the theory can be further extended to more complicated situations, including, for instance, vertical dynamics or higher moments of the strain tensor. However, we stress that horizontal strain and diffusion often are sufficient to describe the evolution of water patches in the ocean, in a relatively wide range of scales and across different regions. This has been experimentally demonstrated in dye release experiments (e.g. Sundermeyer et al. 2020). Now, in the discussion we mention that the model could be extended adding vertical dynamics and/or higher moments of the deformation tensor to address more complex bio-physical processes (line 281). In the supplementary information we provided some results about the sensitivity of the model to different physical parameters finding that the qualitative broad patterns remain unchanged. However, a more extended exploration of spatiotemporal scales could be the focus of a following publication.

Specific comments:

Abstract:

The oxygen budget of the oceans is nearly zero. Better not to mention it here and in the following.
Removed.

I do not fully understand what is a “Lagrangian patch”. In fact, a patch of water is always transported by currents. I would thus avoid adding “Lagrangian” (also later in the text). The Lagrangian one is a theoretical approach and not a physical property, I would say.

We decide to use Lagrangian as an adjective to stress that we follow the evolution of such body of water from a Lagrangian perspective. We would prefer to keep such distinction as is also typically done when “Lagrangian particles” are defined. In our case, the term Lagrangian patch aims to underline the difference with the particle that is usually point-like and shape-less. Note also that, this kind of adjectivation is also used by different authors when defining metrics that depends on trajectories to be calculated, e.g. Lagrangian divergence, vorticity, etc.

“the model resolves”: I would call it the theory and not the model to avoid generating confusion.
Done.

“Propagation” is too vague. Evolution? Change in time?
Done.

Main text:

L4 Why However?

Corrected.

L23 Evolution intended as change in time. Better to be explicit.

Corrected.

L40 Patchiness is here intended as generic heterogeneity. I would use the latter.

Done.

L39 and following: This study is the first that proposes an integrated approach to the effect of diffusion and strain on a plankton patch but it is not the first study that deals with this matter (most of them cited later on in the main text). The point here is the quantification of the impact of heterogeneities on the mean properties. We added the adjective “Lagrangian” to “plankton ecosystems” to be more specific and reduce the generality of the statement. However, we stress that we are not saying that we are the first to study strain and diffusion effects on plankton ecosystems. Instead, we are saying that we found a general mechanism for which spatial variance of plankton distributions can be created just from dilution of a body of water within the surrounding.

In addition, a better justification is needed about its focus on the mean biomass and not on the integrated biomass evolution, the latter being ecologically more relevant, possibly. Indeed, the “observable” is the local biomass concentration and, thus, I agree it should be focus. (In addition, combining the estimate of the mean value with the area of the patch the total biomass is readily obtained, as in the LBA formulations presented later on in the manuscript).

We are sorry but we do not fully understand which quantity we should focus on to justify our assumption. Indeed, we tried to consider the integrated response of a Lagrangian ecosystem in time by defining the LBA that is the time integral of the absolute biomass anomaly. Is it not a sufficiently integrated quantity?

Everywhere in the manuscript: use Ellipse instead of Ellipsis.

Corrected.

L54 Is $S = L + W$ or $(L+W)/2$?

Our definition of characteristic size is $S=L+W$. Because the axes of the ellipse are $2W$ and $2L$. It could also be defined as $S=(L+W)/2$, as long as the quantities that depend on S are consistent with this definition.

L45 and following (including the associated Methods): My understanding is that the theoretical framework is built by first describing the patch as made of a passive tracer (and not a water concentration, as mentioned in the Methods) Theta whose isolines are ellipses, as done in the past for interpreting tracer release experiments. Building upon the equation for the passive tracer evolution an equation is derived to describe the evolution of the patch area (defined using the moments of Theta) as a function of the diffusivity.

We agree with the referee that we did not clearly explain this aspect of our framework. We associate with the patch an ideal concentration of a passive tracer that represents the patch at the initial time, mimicking a situation in which one have dyed all the water molecules in the patch. We stress that this is the same conceptual assumption behind rhodamine or SF6 experiments: identify a physical body of ocean water with a dye concentration. We rewrote the corresponding section in the Methods and we also removed the discussion about the gaussian form of the distribution but that is not a requirement of the theory (lines 305-321).

The latter equation is then used to integrate the equations of the active tracer (the plankton). It is a heuristic approach, evidently, since an active tracer may diffuse differently than a passive one. Therefore, the text describing it has to be clarified since it is currently difficult to grasp the main “conceptual” framework.

We fully agree with the referee that a tracer that undergoes consumption/creation internal processes does not behave as passive. On the other hand, a non-inertial tracer cannot physically disperse and be advected differently from the water in which is dissolved. The key point is to consider the effect of physical processes on the tracer spatial distribution. This is what we did addressing the evolution of means and covariance as function not only of strain and diffusion but also of moments of the tracer distributions present in waters with

different concentrations that are mixed. See also comments below and added references and text in the discussion.

As illustrated in Fig. 3, the diffusion increases the patch. But what is actually “diffusing” is Θ . It is somewhat confusing.

We modified the captions of Fig. 2 and 3 accordingly.

L63 (eq (1)): What is the sensitivity of DA/dt to changes in k and γ , respectively? Why the role of k is explicit, the contribution of γ is actually hidden in the eqs for the evolution of W and L and it is difficult to grasp who is actually regulating the “dilution”.

The strain rate is also responsible for modulating dilution (area increase) because it governs the evolution of the aspect ratio of the patch. Indeed, the stronger is the strain rate the more elongated becomes the patch for a given total area. The quantitative effect of the aspect ratio can be seen in Eq. (1). Indeed we see that $dA/dt \sim (\text{perimeter})^2 / (\text{area})$ because the perimeter of an ellipse $\sim \sqrt{W^2 + L^2}$.

We added the above relation in the main text to clarify this point (line 70).

L66 Can we really state that it is proportional to the perimeter while not having a closed equation of the perimeter value?

We agree that the formula for the perimeter is not exact, but it approximate very well its real value. Indeed, if p is the perimeter we have these upper and lower bounds: $4 \sqrt{W^2 + L^2} \leq p \leq \pi \sqrt{2} \sqrt{W^2 + L^2}$.

L103 and following: Eq. 8 is a bit misleading and also obvious. The novelty here is made of the full equations for mean and variance (which are then integrated numerically) and thus it is necessary to have those equations presented here instead of the equation of plankton local concentrations.

Even if for the referee Eq. (8) is obvious we believe that it would not be so obvious for a general reader of the journal (Nature Communication is a generalist journal) that most likely does not have a specific background of Lagrangian oceanography. That is why we think that Eq. (8) is useful to highlight clearly the main processes acting on a Lagrangian ecosystem. Note also that the equations relative to the evolution of means and covariance are already in the main text, see Eq. (5-6). Finally, Eq. (8) wants also to communicate that the single terms of the equation can be addressed by using different approaches and with different methodologies, leaving space for possible extensions and modification of our framework.

Table 1: It would be better to have values in m instead of km .

We used km since it is a common unit in these kinds of studies, as also reported in other references.

L133 and following: it would be better to present at least one figure (Chl from satellite?) showing the choice of the estimate of the L and W for the real case. There

Figures of the bloom reporting satellite imagery can be found in the cited literature i.e. Ahrham et al. 2000. Note that we used indeed exactly those observed values to initialize the simulation of the SOIREE experiment.

L143-144: Please, define dilution.

Done, we added an explicit definition of dilution in the theory section relating it with the area increase rate (line 73).

L150-151: Is it really surprising to find a relation between heterogeneity and biomass growth, given the non-linear nature of the growth term? In fact, this is the global impact of the effect of “patchiness” studied in the past by Martin, Neufeld etc and it should be discussed here or it should be mentioned that it is further discussed in Section C.

Moreover, is this flow field able to generate patchiness? I would say that heterogeneity is a better term here.

Done, we added those references, also including Mandal and Priyadarshi.

L152: I understand that the dynamical coupling is consider as “implicit” but you could reach a deeper understanding of the system by discussing the variation in time of the values of each term (eg, mean growth, etc) of the equation that describes the “mean” value (

) time evolution, in principle. This exercise could inform also the interpretation of the various sensitivity studies.

We fully agree with the referee. Indeed, in Supplementary Fig. 13 we tried to disentangle different processes (i.e. related to means or covariances) that affect growth. A more exhaustive study would be better addressed by a follow-up publication.

L161: the analysis of the LBA is interesting but it is not an estimate of the total biomass produced. And as demonstrated in the fig. 10, it depends upon the interval of integration (Tau; not explicit in fig. 6 and 7, by the way). To fully understand the observed sensitivity it may useful analyzing also the max of A^ .*

We added explicitly the integration time used in both captions as suggested. We note that, to assess the ecosystem response for very long time, complex grazing and remineralization dynamics should be probably considered. This could be the aim of a future study indeed. However, even considering 1 month of integration we see that the big initial bump of biomass is well captured by our theory and the subsequent relaxation to a near steady (i.e. pre-perturbation) state almost reproduced too. This is consistent with panel d) of Supp. Fig. 10 in which we see that the LBA maxima are not changing dramatically respect to the 30 days simulations.

L236 and following: The Discussion emphasizes the effect of dilution. It would profit a more in depth discussion of the literature (eg, Pasquero's papers mentioned above).

We added references to Pasquero's work (and the seminal work of Plumb), thanks for that (line 282). Pasquero mostly focus on diffusion in a Eulerian model (while our focus is on dilution of a material patch that is also driven by diffusion) and proposes a rescaling of the effective diffusivity to account for non-conservative tracer dynamics in a model where there is no heterogeneity nor tracking of a dilution process happening with external waters with different properties. Pasquero's simulations are based on the work of Plumb (1979) in which eddy fluxes of generic tracers' distributions are derived in a wave-like dynamics. Also in this case, we stress that the approach is Eulerian and advective effects becomes crucial in the definition of the eddy fluxes with an explicit description of tracers' gradients. In our case, since we consider simultaneously (in a time-split fashion) dilution, biological dynamics, and heterogeneity evolution from a true Lagrangian (i.e. material) frame of reference, we do not see an urgent need of introducing an effective diffusivity. Moreover, our approach is different also in the definition of diffusivity itself (rms of velocities deviations vs. of product of velocity deviations with concentration deviations). Finally, we note that using the Plumb framework would be practically almost unfeasible since each tracer should be associated with a time-dependent effective diffusivity complicating the dynamics and adding several unknowns to the system. Nevertheless, we now mentioned the possibility of comparing our approach with models including effective diffusivities in the discussions.

It should also include a discussion about the mechanisms that are known to generate "patchiness" in the oceans, such as filamentation and more generally submesoscales processes. These processes have well defined time and spatial scales and I suspect those scales set specific constraints on the applicability of the approach proposed here. More broadly, a discussion the various mechanisms acting at various scales in the oceans should be briefly discussed here to let the reader understanding in which cases this approach is actually useful (eg, on scales of km or hundreds of km?).

We stress that we describe and model spatial heterogeneity in general with no link to specific oceanographic processes. Our approach is indeed valid in any situation where strain + effective isotropic diffusivity can be regarded as the dominant processes driving mixing and advection. Notably this is a good approximation across a wide range of scales in the ocean (e.g. Ledwell & Sundermeyer works). Then to match a specific scale, the scaling introduced for strain and diffusion can be easily adapted. We added a sentence explaining this and stating that if strain and diffusion are not sufficient, one can use higher moments of the deformation tensor and expand our framework (line 281).

L261: the use of the term "migrate" (coming from neutral ecological theories) is a bit misleading. I quite understand the argument here but it should be clarified that it is intended as transport into the system.

Right, we changed to "dispersal".

L321 and following: *It is an exact derivation but the entire outcome depends upon the definition of A and its evolution in time. In practice, how would A be determined? As an isoline of Chl concentration from satellite data? Or as the evolution of the area based upon the strain (eg, from altimetry) and diffusion rates?*

Very good question. The evaluation of the area could be estimated using, for instance, HF-radar or satellite altimetry. Otherwise, it can be determined a-posteriori using Chl isolines. However, this would mix advection with internal biological dynamics. Using a dye and/or a drifters set could be a more direct approach. We added a sentence about this (line 301).

L364 *The diffusion of an active tracer does follows specific laws since the tracer reacts will diffusing (eg, Pasquero 2015). Please discuss here the assumptions made in eq. 39.*

See our comments above.

Eq. 28. *Is the second term on rhs $_pat$ or $_pat^2$?*

We confirm, up to our knowledge, that the equation is correct as it is.

Fig. 9 Right panel. *The sensitivity analysis is very qualitative and it may be improved by adding a RMS estimate, I would like to suggest. And it shows lower biomass for a higher dilution thus suggesting that the real case is not representative of the biomass enhancement via dilution discussed the in the main text.*

Please note that the biomass concentration is lower but not the LBA. So, the results from SOIREE simulations are consistent with ensemble simulations. We added the Spearman correlation coefficients (fig. 9) that also confirm that the simulation using the observed strain and diffusion is the one that better fit the data but, since we have a few data points available, this conclusion must be taken with caution.

REVIEWERS' COMMENTS

Reviewer #1 (Remarks to the Author):

I am glad to see that the authors have addressed my comments and have been able to use them to make the manuscript even better. I therefore support acceptance of this manuscript

Reviewer #2 (Remarks to the Author):

The authors have addressed mostly of my concerns or satisfactorily argued when not in agreement.

I am happy with the new version of the manuscript.

They have added the Spearman correlations for the data-model comparison. Possibly, given the values are quite few, the significance should be evaluated (or the confidence intervals).

RESPONSE LETTER corresponding to manuscript:

POINT-BY-POINT RESPONSE TO REVIEWER 2

(Referee's comments are reported in *italic* while our answers immediately follow in **red font**.)

hey have added the Spearman correlations for the data-model comparison. Possibly, given the values are quite few, the significance should be evaluated (or the confidence intervals).

We report now in the Figure caption the p-value associated to the Spearman correlation coefficients and we also added a sentence stressing that the number of data points is low.